# Temporal Trends in Air Pollution Exposure across Socioeconomic Groups in The Netherlands

**DOI:** 10.3390/ijerph21080976

**Published:** 2024-07-26

**Authors:** Niklas Hlubek, Yvonne Koop, Alfred Wagtendonk, Ilonca Vaartjes

**Affiliations:** 1Julius Center for Health Sciences and Primary Care, University Medical Center Utrecht, Utrecht University, Internal Mail No. Str. 6.131, P.O. Box 85500, 3508 GA Utrecht, The Netherlands; 2Department of Epidemiology and Data Science, Amsterdam UMC, Vrije Universiteit Amsterdam, De Boelelaan 1117, 1081 HV Amsterdam, The Netherlands

**Keywords:** air pollution, socioeconomic, environmental inequality, longitudinal, PM_2.5_, PM_10_, NO_2_

## Abstract

Air pollution exposure has been linked to detrimental health outcomes. While cross-sectional studies have demonstrated socioeconomic disparities in air pollution exposure, longitudinal evidence on these disparities remains limited. The current study investigates trends in residential air pollution exposure across socioeconomic groups in the Netherlands from 2014 to 2019. Our dataset includes over 12.5 million individuals, aged 18 years and above, who resided in the Netherlands between 2014 and 2019, using Statistics Netherlands data. The address-level air pollution concentrations were estimated by dispersion models of the National Institute of Public Health and the Environment. We linked the exposure estimations of particulate matter < 10 or <2.5 μm (PM_10_, PM_2.5_) and nitrogen dioxide (NO_2_) to household-level socioeconomic data. In highly urbanized areas, individuals from both the lowest and highest socioeconomic groups were exposed to higher air pollution concentrations. Individuals from the lowest socioeconomic group were disproportionally located in highly urbanized and more polluted areas. The air pollution concentrations of PM_10_, PM_2.5_, and NO_2_ decreased between 2014 and 2019 for all the socioeconomic groups. The decrease in the annual average air pollution concentrations was the strongest for the lowest socioeconomic group, although differences in exposure between the socioeconomic groups remain. Further research is needed to define the health and equity implications.

## 1. Introduction

Air pollution is a major health challenge worldwide and is classified by the World Health Organization (WHO) as the largest environmental health risk in Europe [1,2]. Long-term exposure to air pollution such as fine particulate matter (PM_2.5_ and PM_10_) and nitrogen oxide (NO_2_) has consistently been associated with adverse health outcomes, including cardiovascular disease outcomes [3] and mortality [4,5].

Over the last few decades, the majority of the environmental inequality literature from North America, New Zealand, Asia, and Africa has shown that exposure to poor air quality is linked to a lower socioeconomic position (SEP) [6]. Evidence from the European region is rather mixed and the SEP gradient in terms of air pollution exposure is often described as U-shaped, where the lowest SEP group shows the highest levels of exposure, yet also the highest SEP group experiences higher levels of exposure than other groups [7,8,9]. This pattern is likely linked to the spatial distribution of densely populated and highly polluted urban areas, where individuals from the lowest SEP group often reside near pollution sources such as highways or factories, while high SEP individuals often reside in more expensive and polluted city centers [10,11,12].

To protect individuals from air pollution exposure, many initiatives to improve air quality in cities have been set up in the last 30 years. For example, the WHO’s Healthy City Network aims to put health high on the political and social agenda of cities and to build a strong movement for public health at the local level [13]. Furthermore, the adoption and implementation of policy interventions have proved to be effective in improving air quality in North America and Europe by reducing the average population-weighted PM_2.5_ concentrations between 2010 and 2016 in those regions [14,15,16].

Despite the overall improvements in air pollution exposure, SEP-related exposure inequalities in the European region have remained [17,18] or even increased over time in some countries [19,20]. Current evidence on changes in SEP inequalities in air pollution exposure is based on area-level comparisons, missing finer-scale exposure contrasts (e.g., living near a major road). There is a need for longitudinal studies that are carried out at the national level using individual-level data to investigate patterns in the exposure distribution across socioeconomic groups [10,18,21].

To the best of our knowledge, there is no longitudinal study using individual-level data to investigate temporal changes in air pollution exposure across SEP groups on a national level in the European region. The current study aims to fill this gap by investigating nationwide temporal trends in air pollution exposure between 2014 and 2019 in the Netherlands, examining the average annual concentrations of PM_2.5_, PM_10_, and NO_2_ across SEP groups.

## 2. Methods

### 2.1. Study Population and Data Linkage

For this longitudinal study, data on the annual average air pollution concentrations for every year between 2014 and 2019 were linked to sociodemographic data on all the registered residents in the Netherlands. Sociodemographic information was centralized by Statistics Netherlands and originated from the National Population Register, the tax register, and education registers. A detailed description of the sources and generation of the datasets from these registers can be found at www.cbs.nl/microdata (accessed on 23 July 2024).

For every year between 2014 and 2019, we created separate cohorts of all the residents who were registered in the National Population Register in each given year. The average annual air pollution concentrations were then linked to individuals in each cohort based on their residential address data.

We only included individuals who were 18 years or older on 1 January 2014, and did not decease before 31 December 2019. Individuals with missing or incomplete data on air pollution, SEP, or urbanicity in any year were excluded (*n* = 27,911), resulting in a total sample of 12,520,681 individuals who were registered at an address between 2014 and 2019.

### 2.2. Air Pollution Data for the Netherlands, 2014–2019

The average concentrations of PM_2.5_, PM_10_, and NO_2_ for 2014 to 2019 were modeled and mapped on a grid with a resolution of 25 m by the National Institute for Public Health and the Environment (RIVM) and provided by the Geoscience and health Cohort Consortium (GECCO) [22]. The maps were constructed from 1 km resolution nationwide background concentration maps combined with local traffic information. In short, the nationwide background concentration maps were based on dispersion models, including information on industrial, vehicular, and household emissions in the Netherlands and abroad, meteorological data, and chemical information [23]. Two models (one for roads within cities and one for highways in more open terrain) based on local vehicular traffic data, originating from the Dutch National Air Quality Cooperation Program [in Dutch: ‘Nationaal Samenwerkingsprogramma Luchtkwaliteit’] [24,25] were combined with the national background maps [23,25]. The model prediction patterns and absolute concentrations generally agreed well with the measurements for NO_2_ and PM_2.5_ [23,25]; however, quality quantification is hard to interpret as measurements have been used in calibrating the models. Next, the air pollution concentrations were calculated at 9 million datapoints in the Netherlands, which were then interpolated to a raster map with a 25 m resolution using ordinary Kriging. The air pollution concentrations were linked to address locations and exported to a tabular format [22].

### 2.3. Socioeconomic Position

The household-level SEP data were obtained from Statistics Netherlands and hold information on the financial welfare (income and wealth of household), educational level (maximum of main breadwinner and partner), and recent employment history (maximum of main breadwinner and partner) for all the private households in the Netherlands (SES-WOA) [26]. Data were available for all the years from 2014 to 2019. The overall SES-WOA score is a composite of the sub-scores of financial welfare, educational level, and employment history and is calculated by Statistics Netherlands with the use of multiple correspondence analysis [26]. The SEP quintile scores were constructed with the bottom quintile consisting of the lowest SEP (1) and the top quintile representing the highest SEP (5).

### 2.4. Ethnicity

We differentiated between ethnic groups according to Statistics Netherlands’ definition of migration background, which is based on the country of birth of the person and their parents [26]. If the person and both parents were born in the Netherlands, the person’s ethnicity was classified as Dutch. If the person and one or both parents were born abroad (i.e., first-generation migrant), their ethnicity was based on the person’s country of birth. If the person was born in the Netherlands and one of the parents was born abroad (i.e., second-generation migrant), ethnicity was based on the country of birth of the parent born abroad. If the person was born in the Netherlands and both parents were born abroad (i.e., second-generation migrant), ethnicity was based on the mother’s country of birth.

### 2.5. Covariates

Data on age (in years), sex (biological sex as registered at birth), and marital status (married, unmarried and other) were obtained from the Population Register for the years 2014 and 2019 [26]. Neighborhoods in the Netherlands are defined as homogeneously bounded parts of a municipality from a building or socioeconomic perspective [26]. Urbanicity was defined by the neighborhoods’ address density. The neighborhood address density was the average address density per residential address in the neighborhood. The address density per residential address was calculated as the number of addresses in a 1 km^2^ circular buffer around the residential address. We divided urbanicity into two categories, indicating that the residential address was situated in a rural to moderately urbanized neighborhood (<2000 addresses/km^2^) or a highly urbanized neighborhood (>2000 addresses/km^2^).

### 2.6. Statistical Analysis

#### 2.6.1. Summary Statistics

Demographic characteristics were described as the number and percentage of ethnicity (ethnic Dutch, European excluding ethnic Dutch, Indonesian, Turkish. Moroccan and other), marital status (married/unmarried/other), age categories (18–30, 31–40, 41–50, 51–60, 61–70, 71–80, 81+), and SEP group (quintiles) in 2014 and 2019 for the total cohort and stratified by the level of urbanicity (rural to moderately urbanized, highly urbanized). Summary statistics (mean, standard deviation, maximum) were computed for all the air pollutants for all the years from 2014 to 2019 to characterize the annual air pollution concentrations. The annual average air pollution concentrations between 2014 and 2019 were visualized nationwide and stratified by urbanicity.

#### 2.6.2. Average Concentrations across Socioeconomic Position Groups

To examine and visualize the trends in the distribution of air pollution among different SEP groups, we calculated the average annual concentrations of PM_2.5_, PM_10_, and NO_2_ for all the years from 2014 to 2019 per SEP quintiles, stratified by the level of urbanicity. We quantified the absolute changes in the air pollution concentrations between 2014 and 2019 across the SEP quintiles and computed the relative differences to express the changes as a percentage of the concentration in 2014. For a sensitivity analysis, we computed a similar analysis using the SEP deciles (Appendix A).

All the analyses were performed in the secured environment of Statistics Netherlands and performed in R version 4.2.3 (15 March 2023) [27].

## 3. Results

### 3.1. Population Characteristics

Between 2014 and 2019, we identified 12,520,681 individuals aged 18 years or above with registered addresses in the Netherlands. In 2014, we identified 12,000,901 individuals, with 8,371,072 (69.7%) individuals living in rural to moderately urbanized areas and 3,629,829 (30.3%) in highly urbanized areas. In 2019, we identified 11,721,518 individuals, with 7,493,373 (63.9%) living in rural to moderately urbanized areas and 4,228,145 (36.1%) in highly urbanized areas. More individuals with a non-Dutch ethnic background lived in highly urbanized areas (32.2% in 2014; 35.4% in 2019) compared to rural to moderately urbanized areas (14.3% in 2014; 15.7% in 2019). In highly urbanized areas, more unmarried individuals were registered (60.9% in 2014; 60.1% in 2019) compared to rural to moderately urbanized areas, where more married individuals were registered (43.8% in 2014; 43.6% in 2019). Individuals from lower age groups (18–30 and 31–49 years of age) were more often registered in highly urbanized areas. In highly urbanized areas, more than one-quarter of registered individuals were from the lowest SEP group (26.5% in 2014; 26.0% in 2019) compared to 12.0% in 2014 and 14.3% in 2019 in rural to moderately urbanized areas (Table 1).

### 3.2. Annual Average Pollutant Concentrations

Nationally, all the air pollution concentrations decreased between 2014 and 2019. PM_2.5_ showed the largest average concentration decrease (24.2%), followed by PM_10_ (14.6%) and NO_2_ (12.2%). The concentrations of all the air pollutants were consistently lower for individuals living in rural to moderately urbanized areas than in highly urbanized areas. The pattern of change was similar between the categories of urbanicity (Figure 1a–c). Table 2 summarizes the PM_2.5_, PM_10_, and NO_2_ concentrations across all the exposure years.

### 3.3. Differences in Average Air Pollutant Concentrations between 2014 and 2019 by Socioeconomic Position Group

#### 3.3.1. PM_2.5_

Nationwide, the average PM_2.5_ concentrations were higher for the lowest and highest SEP groups compared with the other SEP groups, although the lowest SEP group showed the highest average PM_2.5_ concentrations in 2014 (13.54 µg/m^3^) while the average PM_2.5_ concentrations were similar for both the lowest and the highest SEP group in 2019 (10.23 µg/m^3^; 10.22 µg/m^3^). When stratified for urbanicity, the average PM_2.5_ concentrations in rural to moderately urbanized areas were consistently the highest for the highest SEP group. In highly urbanized areas, the average PM_2.5_ concentrations were higher for the lowest and the highest SEP group, with the highest average concentrations observed in the highest SEP group (Figure 2a).

The relative differences in the average PM_2.5_ concentrations between 2014 and 2019 were higher in highly urbanized areas compared with rural to moderately urbanized areas across the SEP groups. The lower SEP groups showed larger relative differences in the average PM_2.5_ concentrations in highly urbanized areas compared to the other SEP groups. We observed the highest relative difference in the average PM_2.5_ concentrations for the lowest SEP group, both nationwide and in highly urbanized areas. The relative differences in the average PM_2.5_ concentrations across the SEP groups were homogenous in rural to moderately urbanized areas (Table 3).

#### 3.3.2. PM_10_

Nationwide, the average PM_10_ concentrations were higher for the lowest and highest SEP groups compared with the other groups, although the lowest SEP group showed the highest PM_10_ concentrations in 2014 (20.89 µg/m^3^) while the average PM_10_ concentrations were similar for both the lowest and the highest SEP group in 2019 (17.80 µg/m^3^; 17.78 µg/m^3^). When stratified for urbanicity, the average PM_10_ concentrations in rural to moderately urbanized areas were consistently the highest for the highest SEP group. In highly urbanized areas, the average PM_10_ concentrations were consistently higher for the lowest and the highest SEP groups compared with the other groups, with the highest average concentrations in the highest SEP group (Figure 2b).

The relative differences in the average PM_10_ concentrations between 2014 and 2019 were higher in rural to moderately urbanized areas compared with highly urbanized areas across the SEP groups. The lower SEP groups showed larger relative differences in the average PM_10_ concentrations in highly urbanized areas compared to the other groups. We observed the highest relative difference in the average PM_10_ concentrations for the lowest SEP group, both nationwide and across levels of urbanicity (Table 4 and Figure 2b).

#### 3.3.3. NO_2_

Nationwide, the average NO_2_ concentrations were consistently higher for the lowest and highest SEP groups compared with the other groups. Specifically, the lowest SEP group showed the highest NO_2_ concentrations in 2014 (21.53 µg/m^3^) and in 2019 (18.62 µg/m^3^). When stratified for urbanicity, the average NO_2_ concentrations in rural to moderately urbanized areas were consistently the highest for the highest SEP group. In highly urbanized areas, the average NO_2_ concentrations were higher for the lowest and the highest SEP groups compared with the other groups, with the highest average concentrations for the lowest SEP group in 2014 and the highest average concentrations for the highest SEP group in 2019 (Figure 2c).

The relative differences in the average NO_2_ concentrations between 2014 and 2019 were higher in highly urbanized areas compared with rural to moderately urbanized areas across the SEP groups. The lower SEP groups showed larger relative differences in the average NO_2_ concentrations in highly urbanized areas compared to the other groups. We observed the highest relative difference in the average NO_2_ concentrations for the lowest SEP group, both nationwide and across levels of urbanicity (Table 5).

### 3.4. Sensitivity Analysis

We conducted a sensitivity analysis to investigate the potential variations of the differences in the average annual air pollutant concentrations when using the SEP group deciles instead of quintiles. The results of the sensitivity analysis were in line with our initial findings and can be found in Appendix A.

## 4. Discussion

Our study utilizes a comprehensive longitudinal dataset, encompassing over 12.5 million registered individuals in the Netherlands, with detailed demographic and individual-level air pollution data from 2014 to 2019. We present an overview of the temporal changes in exposure to PM_2.5_, PM_10_, and NO_2_ concentrations from 2014 to 2019 across SEP groups.

Between 2014 and 2019, the average air pollution concentrations consistently decreased in the Netherlands. The observed decreases originate from reductions in emissions in various sectors and are driven by autonomous technological development and policy measures. National emission reductions in the Netherlands are required under the Gothenburg Protocol [28] and EU National Emission Ceilings Directive [29]. The largest contributing sector to reductions in emissions contributing to air quality is industry (including electricity productions and refineries), followed by agriculture and transport [30]. The average PM_2.5_ and NO_2_ concentrations showed a stronger decrease in highly urbanized areas, while the average PM_10_ concentrations showed a stronger decrease in rural to moderately urbanized areas. The average PM_2.5_ and PM_10_ concentrations showed a temporal increase from 2017 to 2018. This increase is likely due to changes in the measurement, as municipalities submitted more livestock farms for inclusion in the calculations for 2018 and there were unusual weather conditions in those years, such as a general lack of rain [31].

While the average PM_2.5_, PM_10_, and NO_2_ concentrations in the Netherlands decreased for all the SEP groups between 2014 and 2019, the decrease was generally stronger for the lowest SEP groups. This pattern was more pronounced in highly urbanized areas, whereas in rural to moderately urbanized areas the concentration decreases were more homogenous across the SEP groups. Comparable evidence on the temporal changes in air pollution exposure across SEP groups in the European region is scarce, as most studies are cross-sectional and rely on area-level data. An EU report on air pollution and deprivation from 2007 to 2013/14 indicated that low SEP groups generally benefitted from reductions in air pollution levels at least as much as those in higher SEP groups [18]. Evidence on the national level showed that in Great Britain, the annual average NO_2_ concentrations decreased at lower rates in the most deprived areas between 2001 and 2011 and the annual average PM_10_ concentrations even increased in low SEP areas [20]. Building upon these findings, our study provides more recent evidence, demonstrating a strong decrease in air pollution exposure for individuals in the lowest SEP group between 2014 and 2019 in the Netherlands. This decrease was particularly pronounced in highly urbanized areas, where the main sources of air pollution, such as traffic emissions, are more concentrated. The implementation of environmental and traffic measures, including the banning of older, higher-emission vehicles from densely populated areas and city centers, and the enforcement of speed limits, may have had a significant impact in these areas. This highlights the potential benefits of air pollution reduction strategies for socioeconomically disadvantaged groups. Future research should prioritize the use of individual-level data on a national scale to monitor socioeconomic differences in air pollution changes over time.

Despite the general decrease in the air pollution concentrations between 2014 and 2019 for all the SEP groups, we observed consistent disparities in exposure between the groups, dependent on urbanicity and the type of pollutant. In highly urbanized areas, individuals from both the lowest and highest SEP groups were consistently exposed to higher annual average NO_2_ concentrations compared to the other groups. This U-shaped pattern was similar yet less strong for the annual average PM_2.5_ and PM_10_ concentrations. In rural to moderately urbanized areas, we consistently observed higher annual average air pollution concentrations for the highest SEP group compared to the other SEP groups. Our findings are comparable with a recent Danish study that also used nationwide individual-level data to assess air pollution exposure at the residence [32]. In 2017, Danish individuals from the highest SEP were on average exposed to higher PM_2.5_ and NO_2_ concentrations, whereas in highly urbanized areas, individuals from the lowest SEP group were also exposed the higher average NO_2_ concentrations compared to the other SEP groups. In both the Netherlands and Denmark, individuals from lower SEP groups might relocate to urban areas with higher pollution levels but typically do not reside in the densely trafficked central parts of cities, which are often attractive yet expensive and predominantly inhabited by those from a higher SEP. The housing market conditions in highly urbanized areas may pressure individuals from the lowest SEP group to disproportionately reside near main roads, leading to higher exposure to road transport-related air pollutants such as NO_2_ and PM_2.5_ [33,34]. This could be an explanation for our finding of a U-shaped SEP gradient in highly urbanized areas in the Netherlands, where both the lowest and highest SEP groups are exposed to higher NO_2_ concentrations than other groups.

Although unhealthy levels of air pollution pose a risk to all individuals, those from lower SEP groups may exhibit increased susceptibility to the associated health risks [35]. This heightened vulnerability could stem from pre-existing health conditions or other harmful exposures associated with a lower social position, such as occupational hazards, which could further impair their health status [10]. Our study demonstrates that individuals from the lowest SEP group were disproportionately located in highly urbanized and heavily polluted areas in the Netherlands, representing the largest group in these areas compared to any other SEP group. The health risks associated with air pollution may be particularly relevant for this more exposed and vulnerable population group.

Among the strengths of the current study are that it benefitted from data on the entire population of the Netherlands aged 18 and above between 2014 and 2019, with the usage of nationwide socioeconomic data on the household level and the modeling of the air pollution concentration at the residential address-level. While the majority of studies on environmental inequality are carried out cross-sectionally, the current study contributes to the understanding of temporal changes in air pollution exposure across SEP groups. We were able to compute the results on a population-wide dataset providing a comprehensive portrayal of the changing social distribution of air pollution exposure between 2014 and 2019.

The limitations include the fact that the current study only focused on exposure to PM_2.5_, PM_10_ and NO_2_ concentrations and the associations might differ for other air pollutants, such as ultrafine particles and black carbon. Furthermore, the modeled air pollution concentrations at the household level can only be seen as a proxy for personal exposure. Although the air pollution concentrations from the prediction models showed good agreement [23,25], we acknowledge some remaining uncertainty; however, by averaging over large-scale population groups, over- or underestimations are minimized, ensuring the robustness of our results. Actual exposure is also influenced by indoor air pollution, exposure during occupation, and mobility patterns. Additionally, exposure variations exist at the street-level scale, influenced by factors such as the altitude of the living floor and the distance from the façade to the street center [36]. The associations between the sociodemographic characteristics of the population and the residential air pollution concentrations are likely location-specific and generalization to other countries should be performed with caution.

## 5. Conclusions

The air pollution concentrations (PM_2.5_, PM_10_, and NO_2_) have decreased for all the SEP groups in the Netherlands between 2014 and 2019, with the strongest decrease for the lowest SEP group. The average PM_2.5_ and NO_2_ concentrations showed a stronger decrease in highly urbanized areas, while the average PM_10_ concentrations showed a stronger decrease in rural to moderately urbanized areas. Within rural to moderate urbanized areas, individuals in the highest SEP group were consistently exposed to the highest air pollution concentrations. In highly urbanized areas, individuals from the highest and lowest SEP group were exposed to higher air pollution concentrations compared to the other SEP groups. Individuals from the lowest SEP group were disproportionately located in highly urbanized and polluted areas compared to any other SEP group. Our findings highlight that while exposure to air pollution decreased more in favor of the lowest SEP group in the Netherlands between 2014 and 2019, the average exposure differences between socioeconomic groups persist and vary depending on the specific pollutant and urbanization level.

## Figures and Tables

**Figure 1 ijerph-21-00976-f001:**
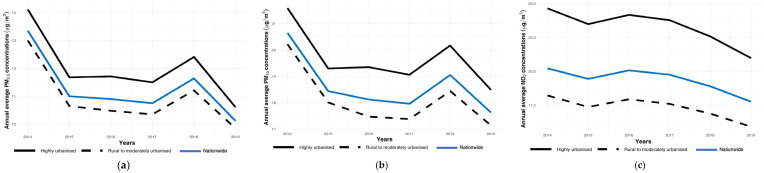
(**a**) Annual average PM_2.5_ concentrations in the Netherlands for the period 2014–2019 by urbanicity. Annual average PM_2.5_ concentrations (µg/m^3^); nationwide and by urbanicity (rural to moderately urbanized and highly urbanized areas) (**b**) Annual average PM_10_ concentrations in the Netherlands for the period 2014–2019 by urbanicity. Annual average PM_10_ concentrations (µg/m^3^); nationwide and by urbanicity (rural to moderately urbanized and highly urbanized areas). (**c**) Annual average NO_2_ concentrations in the Netherlands for the period 2014–2019 by urbanicity. Annual average NO_2_ concentrations (µg/m^3^); nationwide and by urbanicity (rural to moderately urbanized and highly urbanized areas).

**Figure 2 ijerph-21-00976-f002:**
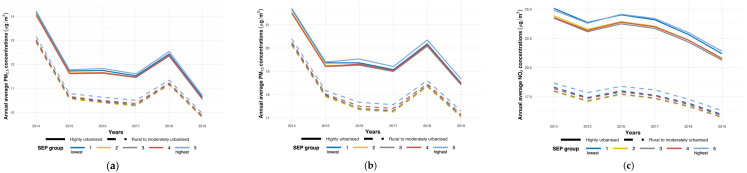
(**a**) Annual average PM_2.5_ concentrations in the Netherlands for the period 2014–2019 by urbanicity and SEP group. Annual average PM_2.5_ concentrations (µg/m^3^) by urbanicity (rural to moderately urbanized and highly urbanized areas) and socioeconomic position (SEP) group (1—lowest to 5—highest). (**b**) Annual average PM_10_ concentrations in the Netherlands for the period 2014–2019 by urbanicity and SEP group. Annual average PM_10_ concentrations (µg/m^3^) by urbanicity (rural to moderately urbanized and highly urbanized areas) and socioeconomic position (SEP) group (1—lowest to 5—highest). (**c**) Annual average NO_2_ concentrations in the Netherlands for the period 2014–2019 by urbanicity and SEP group. Annual average NO_2_ concentrations (µg/m^3^) by urbanicity (rural to moderately urbanized and highly urbanized areas) and socioeconomic position (SEP) group (1—lowest to 5—highest).

**Table 1 ijerph-21-00976-t001:** Demographic characteristics of registered individuals in the Netherlands for 2014 and 2019.

	Nationwide		Rural to Moderately Urbanized Areas ^1^		Highly Urbanized Areas ^2^	
	2014	2019	2014	2019	2014	2019
**N (%)**	12,000,901 (100)	11,721,518 (100)	8,371,072 (69.7)	7,493,373 (63.9)	3,629,829 (30.3)	4,228,145 (36.1)
**Female—N (%)**	6,123,174 (51.0)	5,971,023 (50.9)	4,247,354 (50.7)	3,805,917 (50.8)	1,875,820 (51.7)	2,165,106 (51.2)
**Ethnicity—N (%)**						
Ethnic Dutch	9,592,811 (79.9)	9,036,258 (77.1)	7,170,794 (85.7)	6,305,021 (84.1)	2,422,017 (66.7)	2,731,237 (64.6)
European excl. Dutch	755,728 (6.3)	888,467 (7.6)	464,734 (5.6)	474,209 (6.3)	290,994 (8.0)	414,258 (9.8)
Indonesian	318,185 (2.7)	310,942 (2.7)	195,601 (2.3)	175,277 (2.3)	122,584 (3.4)	135,665 (3.2)
Turkish	263,169 (2.2)	267,702 (2.3)	102,451 (1.2)	90,932 (1.2)	160,718 (4.4)	176,770 (4.2)
Surinamese	252,579 (2.1)	252,185 (2.2)	89,988 (1.1)	84,651 (1.1)	162,591 (4.5)	167,534 (4.0)
Moroccan	228,745 (1.9)	232,732 (2.0)	78,287 (0.9)	70,253 (0.9)	150,458 (4.1)	162,479 (3.8)
Other	589,683 (4.9)	733,131 (6.4)	269,216 (3.2)	293,029 (3.9)	320,467 (8.8)	440,202 (10.4)
**Marital status—N (%)**						
Married	6,122,843 (51.0)	5,910,200 (50.4)	4,705,716 (56.2)	4,223,459 (56.4)	1,417,127 (39.0)	1,686,741 (39.9)
Unmarried	4,029,771 (33.6)	3,522,039 (30.0)	2,415,847 (28.9)	1,822,170 (24.3)	1,613,924 (44.5)	1,699,869 (40.2)
Other	1,848,287 (15.4)	2,289,279 (19.5)	1,249,509 (14.9)	1,447,744 (19.3)	598,778 (16.4)	841,535 (19.9)
**Age in years—N (%)**						
18–30	2,286,146 (19.4)	1,598,208 (13.6)	1,386,812 (16.8)	811,930 (10.8)	899,334 (25.1)	786,278 (18.6)
31–40	1,908,844 (16.2)	1,944,693 (16.6)	1,236,182 (15.0)	1,139,251 (15.2)	672,662 (18.8)	805,442 (19.0)
41–50	2,423,299 (20.5)	2,103,797 (17.9)	1,768,591 (21.5)	1,388,762 (18.5)	654,708 (18.3)	715,035 (16.9)
51–60	2,225,377 (18.8)	2,291,276 (19.5)	1,642,871 (20.0)	1,572,887 (21.0)	582,506 (16.3)	718,389 (17.0)
61–70	1,811,973 (15.3)	1,886,903 (16.1)	1,349,278 (16.4)	1,295,481 (17.3)	462,695 (12.9)	591,422 (14.0)
71–80	910,492 (7.7)	1,322,422 (11.3)	673,761 (8.2)	908,605 (12.1)	236,731 (6.6)	413,817 (9.8)
81+	247,790 (2.1)	574,219 (4.9)	173,734 (2.1)	376,457 (5.0)	74,056 (2.1)	197,762 (4.7)
**SEP Quintiles—N (%)**						
1—lowest	2,040,311 (17.0)	2,171,763 (18.5)	1,078,572 (12.9)	1,071,003 (14.3)	961,739 (26.5)	1,100,760 (26.0)
2	2,503,730 (20.9)	2,456,233 (21.0)	1,741,261 (20.8)	1,573,113 (21.0)	762,469 (21.0)	883,120 (20.9)
3	2,539,294 (21.2)	2,413,635 (20.6)	1,910,958 (22.8)	1,669,345 (22.3)	628,336 (17.3)	744,290 (17.6)
4	2,476,620 (20.6)	2,372,348 (20.2)	1,896,889 (22.7)	1,679,045 (22.4)	579,731 (16.0)	693,303 (16.4)
5—highest	2,440,946 (20.3)	2,307,539 (19.7)	1,743,392 (20.8)	1,500,867 (20.0)	697,554 (19.2)	806,672 (19.1)

^1^ Rural to moderately urbanized areas (<2000 addresses/km^2^). ^2^ Highly urbanized areas (>2000 addresses/km^2^).

**Table 2 ijerph-21-00976-t002:** Annual air pollution concentration for PM_2.5_, PM_10_, and NO_2_ in the Netherlands for the period 2014 and 2019, nationwide and by urbanicity.

Year	Urbanicity ^1^	PM_2.5_ (µg/m^3^)—Mean (SD)	PM_2.5_ (µg/m^3^)—Max	PM_10_ (µg/m^3^)—Mean (SD)	PM_10_ (µg/m^3^)—Max	NO_2_ (µg/m^3^)—Mean (SD)	NO_2_ (µg/m^3^)—Max
2014	Nationwide	13.35 (1.42)	19.53	20.65 (1.70)	40.66	20.22 (5.17)	51.20
	Highly urbanized	14.11 (1.11)	17.70	21.59 (1.41)	27.42	24.64 (4.44)	51.20
	Rural to moderately urbanized	13.00 (1.31)	19.53	20.22 (1.78)	40.66	18.22 (4.12)	45.89
2015	Nationwide	11.00 (1.37)	26.91	18.44 (1.73)	36.85	19.44 (5.15)	49.55
	Highly urbanized	11.68 (1.11)	26.91	19.30 (1.38)	35.16	23.48 (4.39)	47.89
	Rural to moderately urbanized	10.65 (1.36)	26.50	18.01 (1.72)	36.85	17.37 (4.19)	49.55
2016	Nationwide	10.90 (1.48)	17.51	18.12 (1.82)	35.50	20.07 (5.07)	60.13
	Highly urbanized	11.71 (1.17)	16.94	19.35 (1.48)	25.80	24.16 (4.31)	60.13
	Rural to moderately urbanized	10.48 (1.46)	17.51	17.47 (1.64)	35.50	17.94 (4.03)	45.97
2017	Nationwide	10.75 (1.38)	16.62	17.96 (1.76)	30.34	19.75 (5.21)	45.35
	Highly urbanized	11.50 (1.07)	16.32	19.06 (1.40)	25.65	23.77 (4.43)	45.35
	Rural to moderately urbanized	10.35 (1.37)	16.62	17.38 (1.65)	30.34	17.60 (4.24)	42.04
2018	Nationwide	11.64 (1.29)	17.29	19.05 (1.65)	31.03	18.89 (4.71)	43.83
	Highly urbanized	12.41 (1.04)	17.05	20.17 (1.36)	26.70	22.57 (4.12)	43.33
	Rural to moderately urbanized	11.21 (1.21)	17.29	18.44 (1.46)	31.03	16.88 (3.68)	43.83
2019	Nationwide	10.12 (1.08)	15.63	17.63 (1.43)	29.13	17.76 (4.26)	42.43
	Highly urbanized	10.61 (0.91)	13.39	18.44 (1.19)	23.52	20.97 (3.74)	40.88
	Rural to moderately urbanized	9.85 (1.08)	15.63	17.14 (1.32)	29.13	15.92 (3.34)	42.43

^1^ Urbanicity: rural to moderately urbanized areas (<2000 addresses/km^2^); highly urbanized areas (>2000 addresses/km^2^).

**Table 3 ijerph-21-00976-t003:** Differences in the average PM_2.5_ concentrations between 2014 and 2019 by socioeconomic position group and urbanicity (2014 to 2019).

Urbanicity ^1^	SEP Group ^2^	PM_2.5_ (µg/m^3^)—Mean 2014	PM_2.5_ (µg/m^3^)—Mean 2019	Absolute Difference	Relative Difference (%)
Nationwide	1	13.54	10.23	−3.3	−24.5
	2	13.27	10.07	−3.2	−24.1
	3	13.22	10.04	−3.2	−24.0
	4	13.26	10.07	−3.2	−24.1
	5	13.46	10.22	−3.2	−24.1
Rural to moderately urbanized	1	12.98	9.82	−3.2	−24.3
	2	12.91	9.79	−3.1	−24.2
	3	12.94	9.81	−3.1	−24.2
	4	13.01	9.86	−3.2	−24.2
	5	13.14	9.96	−3.2	−24.3
Highly urbanized	1	14.17	10.63	−3.5	−25.0
	2	14.07	10.57	−3.5	−24.9
	3	14.04	10.55	−3.5	−24.9
	4	14.05	10.57	−3.5	−24.8
	5	14.21	10.71	−3.5	−24.6

^1^ Urbanicity: rural to moderately urbanized areas (<2000 addresses/km^2^); highly urbanized (>2000 addresses/km^2^). ^2^ Socioeconomic position (SEP) group: quintile scores ranging from the lowest SEP (1) to the highest SEP (5).

**Table 4 ijerph-21-00976-t004:** Differences in the average PM_10_ concentrations between 2014 and 2019 by socioeconomic position group and urbanicity (2014 to 2019).

Urbanicity ^1^	SEP Group ^2^	PM_10_ (µg/m^3^)—Mean 2014	PM_10_ (µg/m^3^)—Mean 2019	Absolute Difference	Relative Difference (%)
Nationwide	1	20.89	17.80	−3.1	−14.8
	2	20.56	17.55	−3.0	−14.6
	3	20.49	17.50	−3.0	−14.6
	4	20.55	17.55	−3.0	−14.6
	5	20.78	17.78	−3.0	−14.4
Rural to moderately urbanized	1	20.19	17.08	−3.1	−15.4
	2	20.11	17.04	−3.1	−15.2
	3	20.15	17.09	−3.1	−15.2
	4	20.24	17.16	−3.1	−15.2
	5	20.39	17.30	−3.1	−15.2
Highly urbanized	1	21.67	18.50	−3.2	−14.6
	2	21.54	18.42	−3.1	−14.5
	3	21.49	18.41	−3.1	−14.3
	4	21.49	18.47	−3.1	−14.1
	5	21.70	18.67	−3.0	−14.0

^1^ Urbanicity: rural to moderately urbanized areas (<2000 addresses/km^2^); highly urbanized (>2000 addresses/km^2^). ^2^ Socioeconomic position (SEP) group: quintile scores ranging from the lowest SEP (1) to the highest SEP (5).

**Table 5 ijerph-21-00976-t005:** Differences in the average NO_2_ concentrations between 2014 and 2019 by socioeconomic position group and urbanicity (2014 to 2019).

Urbanicity ^1^	SEP Group ^2^	NO_2_ (µg/m^3^)—Mean 2014	NO_2_ (µg/m^3^)—Mean 2019	Absolute Difference	Relative Difference (%)
Nationwide	1	21.53	18.62	−2.9	−13.5
	2	19.97	17.54	−2.4	−12.2
	3	19.59	17.27	−2.3	−11.8
	4	19.68	17.34	−2.3	−11.8
	5	20.52	18.11	−2.4	−11.7
Rural to moderately urbanized	1	18.32	15.98	−2.4	−12.8
	2	17.95	15.71	−2.3	−12.5
	3	18.01	15.75	−2.3	−12.5
	4	18.20	15.89	−2.3	−12.7
	5	18.66	16.31	−2.4	−12.6
Highly urbanized	1	25.07	21.18	−3.9	−15.5
	2	24.43	20.74	−3.7	−15.1
	3	24.23	20.62	−3.6	−14.9
	4	24.29	20.79	−3.5	−14.4
	5	24.91	21.40	−3.5	−14.1

^1^ Urbanicity: rural to moderately urbanized areas (<2000 addresses/km^2^); highly urbanized (>2000 addresses/km^2^). ^2^ Socioeconomic position (SEP) group: quintile scores ranging from the lowest SEP (1) to the highest SEP (5).

## Data Availability

The results are based on calculations using geodata and non-public microdata from Statistics Netherlands; the full datasets are not available due to privacy regulations. Under certain conditions, the underlying encrypted microdata are accessible for statistical and scientific research. For further information, contact microdata@cbs.nl. If verification of the analyses is desired and Statistics Netherlands provides access to the microdata, we will provide the R-scripts for the cohort-building and analyses upon request to the corresponding author. The geodata can be downloaded from www.atlasleefomgeving.nl or requested from the Geoscience and Health Cohort Consortium (www.gecco.nl).

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
