# Peer review of "Temporal Trends in Air Pollution Exposure across Socioeconomic Groups in The Netherlands"

_ijerph, 2024, doi:10.3390/ijerph21080976_

Round 1

Reviewer 1 Report

Comments and Suggestions for Authors

This is a well written paper with a large data set being used.

The results are probably what one would have expected, as the more polluted areas have more scope to improve. 

In some instances the differences between SE are actually quite small.

What would have really made this work stand out is if another year of data were to be included (2020) as this would add in the effects of COVID, and greatly increase the potential impact of this work.

Author Response

Response to Reviewer Comments

Reviewer 1 Comments:

Comment 1:

“The results are probably what one would have expected, as the more polluted areas have more scope to improve.”

Response 1:

Thank you for your comment. While it may seem intuitive that more polluted areas would show greater improvement over time, it is important to note that there is actually limited empirical evidence on the socioeconomic distribution and temporal changes of those areas. The number of studies examining longitudinal changes in socioeconomic disparities in air pollution exposure is very limited and evidence is mixed. For example, between 2001 and 2011, in the UK, average NO2 concentrations have fallen, yet the rate of improvement has been slower for more deprived areas and average PM10 concentrations have risen and done so more quickly for the poor (Mitchell et al., 2015). Longitudinal changes in socioeconomic disparities in air pollution exposure have been examined on a neighbourhood level on the EU level (EEA, 2018), yet to our knowledge there are no recent nationwide studies using individual level socioecnomic data. The current study therefore extends the existing evidence in the field.

European Environment Agency. (2018). Analysis of Air Pollution and Noise and Social Deprivation. Retrieved from https://www.eionet.europa.eu/etcs/etc-atni/products/etc-atni-reports/eionet_rep_etcacm_2018_7_deprivation_aq_noise

Mitchell, G., Norman, P., & Mullin, K. (2015). Who benefits from environmental policy? An environmental justice analysis of air quality change in Britain, 2001–2011. Environmental Research Letters, 10(10).https://doi.org/10.1088/1748 9326/10/10/105009

Comment 2:

“In some instances the differences between SE are actually quite small.”

Response 2:

Thank you for your comment. It is important to consider that our study utilized nationwide data encompassing more than 12.5 million individuals to calculate large-scale annual average air pollution concentrations across population groups. By selecting socioeconomic quintiles, we aim to offer a balanced analysis of disparities in air pollution exposure, ensuring sufficient detail without excessive fragmentation into numerous smaller groups. Although some differences in exposure between socioeconomic groups may appear small, they represent significant disparities impacting large groups of individuals. On a population level, these differences can have substantial implications for public health. We believe this approach provides a robust overview of temporal changes in socioeconomic disparities in air pollution exposure. In our next study, we will examine socioeconomic health inequalities associated with long-term air pollution exposure.

Comment 3: 

“What would have really made this work stand out is if another year of data were to be included (2020) as this would add in the effects of COVID, and greatly increase the potential impact of this work.”

Response 3:

Thank you for your suggestion regarding the inclusion of 2020 data to account for the effects of COVID-19. We understand that incorporating this additional year could potentially enhance the impact of our study by capturing the unique changes in air pollution exposure due to the pandemic. However, several considerations influenced our decision to exclude 2020 in this initial analysis. The COVID-19 pandemic not only altered air pollution levels but also had profound effects on the economy, healthcare system, and human behavior. These changes introduce multiple confounding variables for which we currently do not have comprehensive data, necessary to accurately account for their impacts.

We specifically designed this study as a precursor to a follow-up study that will include health outcomes. Including 2020 data from a health perspective could complicate the analysis due to the pandemic's multifaceted effects on health and healthcare access, making it challenging to isolate the impacts of air pollution. Additionally, the individual-level socioeconomic data was not available when we designed the study and carried out the initial analysis. This data only recently became available, and adding it now would be both costly and time-intensive. While adding the 2020 data could provide valuable insights, the complexity and current limitations in available data guided our decision to exclude it from this preliminary analysis.

Reviewer 2 Report

Comments and Suggestions for Authors

Exposure to air pollution by socioeconomic groups is the primary focus of this paper. The authors used averaging techniques to calculate air pollution fields in the Netherlands with fine resolution (25m) using dispersion and averaging models. The first criticism arises from the use of these models because the results can be affected by uncertainty. The differences in exposition between the SEP groups and the different locations were relatively small. Authors are invited to discuss this point, especially in ch.2.2. I also call the attention of the authors to Tables, where the concentrations are shown with four significant digits,

In addition, air pollution follows specific patterns depending on the source intensity, distribution, chemical reactions, local and regional meteorology, and so on. Therefore, the association between groups with different SEP groups depends on the distribution of the population in the Country. A reader not very familiar with this subject may arrive at an incorrect socioeconomic conclusion on low SEP discrimination, as evidenced in the last sentence of the conclusions.

In ch 3.2, a discussion on the evolution of air pollution between 2014 and 2019 is given. All pollutants exhibit a nearly marked decrease in concentration; are these results the result of specific air pollution control regulations? The Authors are invited to provide explanations.

In Figures 1-3, increase the font size.

Author Response

Response to Reviewer Comments

Reviewer 2 Comments:

Comment 1:

“The authors used averaging techniques to calculate air pollution fields in the Netherlands with fine resolution (25m) using dispersion and averaging models. The first criticism arises from the use of these models because the results can be affected by uncertainty. The differences in exposition between the SEP groups and the different locations were relatively small. Authors are invited to discuss this point, especially in ch.2.2.” 

Response 1:

Thank you for your comment. The data we used is sourced from the National Institute for Public Health and the Environment (RIVM), which used 'Ordinary Kriging' technique to spread estimated air pollution concentrations at the home level over a fine resolution of 25x25 meters, in contrast to other models that typically use a coarser 1x1 km scale (Velders et al., 2011). The prediction patterns and uncertainties of these models have been evaluated and show good agreement with observed concentrations (Velders et al., 2011; Wesseling et al., 2018). We have added a paragraph in section 2.2 to highlight this:

Model prediction patterns and absolute concentrations generally agreed well with measurements for NO2 and PM2.5, (Velders et al. 2011; Wesseling et al., 2018), however, quality quantification is hard to interpret as measurements have been used in calibrating the models.

While acknowledging model uncertainty, our results represent large-scale nationwide averages of individual address-level estimates across population groups, ensuring that any over or underestimations are effectively averaged out. Even though differences in exposure between SEP groups and different locations may appear small, these differences can have significant public health implications when viewed across large populations. We have also added a paragraph to the limitation section of the article to make this clearer:

Although air pollution concentrations from the prediction models showed good agreement (Velders et al., 2011; Wesseling et al., 2018), we acknowledge some remaining uncertainty; however, by averaging over large-scale population groups, over or underestimations are minimized, ensuring the robustness of our results.

We are certain that our analysis approach provides a robust overview of temporal changes in socioeconomic disparities in air pollution exposure.

Velders, G. J. M., Aben, J. M. M., Jimmink, B. A., van der Swaluw, E., de Vries, W. J., Geilenkirchen, G. P., & Matthijsen, J. (2011). Large-scale Air Quality Concentration and Deposition Maps in the Netherlands—Report 2011. RIVM Re-port, 680362001.

Wesseling, J., Nguyen, L., & Hoogerbrugge, R. (2018). Gemeten en berekende concentraties stikstof(di)oxiden en fijnstof in de periode 2010 t/m 2015 (Update): Een test van de standaardrekenmethoden 1 en 2. https://doi.org/10.21945/RIVM-2016-0106

Comment 2:

“I also call the attention of the authors to Tables, where the concentrations are shown with four significant digits.”

Response 2:

Thank you for pointing this out. We have reviewed the tables in the uploaded file to ensure that the concentration values are presented correctly.

Comment 3:

“In addition, air pollution follows specific patterns depending on the source intensity, distribution, chemical reactions, local and regional meteorology, and so on. Therefore, the association between groups with different SEP groups depends on the distribution of the population in the country. A reader not very familiar with this subject may arrive at an incorrect socioeconomic conclusion on low SEP discrimination, as evidenced in the last sentence of the conclusions.”

Response 3:

Thank you for your comment. We agree that air pollution distribution is influenced by a variety of factors which combined with the population distribution in the country, play a significant role in determining the exposure levels of different socioeconomic groups. Due to housing market conditions, individuals from low SEP groups may be disproportionately located close to main roads in highly urbanised areas with high volumes of road transport related air pollution exposure (e.g., NO2 and PM2.5) (Mielck & Heinrich, 2002; EEA, 2019). Results from the current study provide evidence for a U-shaped distribution of air pollution exposure in urban areas in the Netherlands, where both individuals from the lowest and highest SEP group are exposed to higher concentrations in those highly urbanised areas. Importantly, we find that individuals from the lower SEP groups are disproportionately located in those highly urbanised and more polluted areas. We have added paragraphs to relevant parts of section 4 to clarify the relevance of population distribution in highly urbanised and heavily polluted areas:

In (…) the Netherlands (…), individuals from lower SEP groups might relocate to urban areas with higher pollution levels but typically don’t reside in the densely trafficked central parts of cities, which are often attractive yet expensive and predominantly inhabited by those with higher SEP. Housing market conditions in highly urbanized areas may pressure individuals from the lowest SEP group to disproportionately reside near main roads, leading to higher exposure to road transport related air pollutants such as NO2 and PM2.5 (Mielck & Heinrich, 2002; EEA, 2019). This could be an explanation of our results for a U-shaped SEP gradient in highly urbanised areas in the Netherlands where both the lowest and highest SEP groups are exposed to higher NO2 concentrations than other groups.

Furthermore, we also altered the phrasing in the last sentence in section 5 to avoid incorrect conclusions about low SEP discrimination:

Our findings highlight that while exposure to air pollution decreased more in favour of the lowest SEP group in the Netherlands between 2014 and 2019, average exposure differences between socioeconomic groups persist and vary depending on the specific pollutant and urbanisation level.

European Environment Agency. Air Quality in Europe. 2019 report; Publications Office of the European Union: Luxembourg, 2019; ISBN 978-92-9480-088-6.

Mielck, A. & Heinrich, J., (2002). Social inequalities and distribution of the environmental burden among the population (environmental justice). Gesundheitswesen (Bundesverband der Arzte des Offentlichen Gesundheitsdienstes (Germany)). 2002 Jul;64(7):405-416. https://doi.org/10.1055/s-2002-32815

Comment 4:

“In ch 3.2, a discussion on the evolution of air pollution between 2014 and 2019 is given. All pollutants exhibit a nearly marked decrease in concentration; are these results the result of specific air pollution control regulations? The Authors are invited to provide explanations.”

Response 4:

Thank you for your comment. Air quality in the Netherlands has improved since 1980 due to European policies. The decrease in concentrations between 2014 and 2019 can not be attributed to specific air pollution control regulations, but to autonomous technological developments and/or policy measures such as the Gotheburg protocol (UNECE, 1999) and EU national Emission Ceilings directive (EU, 2001). In section  4, we have added a paragraph to highlight which policy measures are related to the reductions in air pollution concentrations and which sectors contributed the most to these reductions:

The observed decreases originate from reductions in emission in various sectors and are driven by autonomous technological development and policy measures. National emission reductions in the Netherlands are obliged under the Gotheburg protocol (UNECE, 1999) and EU national Emission Ceilings directive (EU, 2001). The largest contributing sector to reductions in emissions contributing to air quality is industry (including electricity productions and refineries), followed by agriculture and transport (Velders et al., 2020).

UNECE. (1999). Protocol to the 1979 Convention on long-range transboundary air pollution to abate acidification, eutrophication and ground-level ozone. UN/ECE Document EB/AIR/1999/1

EU. (2001). Directive 2001/81/EC of the European Parliament and of the Council of 23 October 2001 on the National Emissions Ceilings for Certain Atmospheric Pollutants. European Commission, Brussels, Belgium (2001).

Comment 5:

“In Figures 1-3, increase the font size.”

Response 5:

Thank you for your suggestion. We have increased the font size for Figures 1a-c and 2a-c to improve readability.